

# On the role of tropopause folds in summertime tropospheric ozone over the eastern Mediterranean and the Middle East

Dimitris Akritidis[1,2], Andrea Pozzer[1], Prodromos Zanis[2], Evangelos Tyrlis[3], Bojan Škerlak[4], Michael Sprenger[4], and Jos Lelieveld[1,3]

[1]Max Planck Institute for Chemistry, Mainz, Germany
[2]Department of Meteorology and Climatology, School of Geology, Aristotle University of Thessaloniki, Thessaloniki, Greece
[3]Energy, Environment and Water Research Center, The Cyprus Institute, Nicosia, Cyprus
[4]Institute for Atmospheric and Climate Science, ETH Zurich, Zurich, Switzerland

*Correspondence to:* D. Akritidis (dakritid@geo.auth.gr)

**Abstract.** We study the contribution of tropopause folds in the summertime pool of tropospheric ozone over the eastern Mediterranean and the Middle East (EMME) with the aid of the atmospheric chemistry climate model ECHAM5/MESSy (EMAC). Tropopause fold events in EMAC simulations were identified with a 3-D labeling algorithm that detects folds at grid points where multiple crossings of the dynamical tropopause are computed. Subsequently the events featuring the largest

horizontal and vertical extent were selected for further study. For the selection of these events we identified a significant contribution of the stratospheric ozone reservoir to the high concentrations of ozone in the middle/lower free troposphere over the EMME. A distinct increase of ozone is found over the EMME in the middle troposphere during summer as a result of the fold activity, shifting towards the south-east and decreasing altitude. We find that the interannual variability of near surface ozone over the eastern Mediterranean (EM) during summer is related to that of both tropopause folds and ozone in the free

troposphere.

## 1   Introduction

Tropospheric ozone is a key species controlling the oxidation capacity of the troposphere (Crutzen, 1988; Penkett, 1988), while it acts as a greenhouse gas in terms of radiative forcing at the earth's surface (Solomon et al., 2007). Compared to ozone near the surface, ozone in the free troposphere can be transported over greater distances due to its relatively longer lifetime and the

higher wind velocities, while owing to the high radiative forcing efficiency in the upper troposphere has proportionally greater impact on climate (Lacis et al., 1990). Tropospheric ozone originates from both chemical production through a sequence of reactions from its precursors (nitrogen oxide, volatile organic compounds and carbon monoxide) in the presence of sunlight (Crutzen, 1974) and downward transport from the stratosphere (Danielsen, 1968). Although the abundance and distribution of tropospheric ozone are mainly controlled by photochemistry (Lelieveld and Dentener, 2000), the relative contribution of

stratospheric ozone to the tropopsheric ozone budget can be significant in certain regions (Roelofs and Lelieveld, 1997; Zanis et al., 2014).



The EM (approximately 20-35°E and 30-45°N) basin is a region of great interest as it is associated with one of the highest levels of background tropospheric ozone around the globe (Zerefos et al., 2002). During summer, the region is characterized by cloud-free conditions and high solar radiation intensity, which along with the polluted air masses arriving from Europe, Africa and Asia (Lelieveld et al., 2002; Kanakidou et al., 2011), result in enhanced photochemical production of ozone. Therefore,

air quality standards of the European Union are often violated (Kouvarakis et al., 2002), potentially having a strong impact on regional air quality and climate (Hauglustaine and Brasseur, 2001). Moreover, the summertime circulation over EM favours the downward transport throughout the depth of the troposphere (Ziv et al., 2004; Tyrlis et al., 2013; Zanis et al., 2014), while a global hot spot of tropopause fold formation has been identified over the area (Sprenger et al., 2003; Traub and Lelieveld, 2003; Tyrlis et al., 2014; Škerlak et al., 2015).

During recent years many observational studies have focused on the marked enhancement of summertime ozone over the EM, involving analysis of measurement data from rural and baseline stations (Kouvarakis et al., 2000; Kalabokas and Repapis, 2004; Gerasopoulos et al., 2005), field campaigns (Kourtidis et al., 2002; Kalabokas et al., 2013; Tombrou et al., 2015) and satellites (Richards et al., 2013; Doche et al., 2014; Safieddine et al., 2014). There is also a number of modeling studies on the summertime tropospheric ozone buildup over the EMME (approximately 20-50°E and 20-45°N) in an attempt to unravel

the contributing mechanisms (Zerefos et al., 2002; Lelieveld et al., 2009; Liu et al., 2009, 2011; Zanis et al., 2014). Zerefos et al. (2002) showed that the high ozone levels over the EM cannot be controlled through local emissions. Instead, they are mostly influenced by long-range import of air masses, rich in ozone and ozone precursors, from the European continent (in the lower troposphere) (Stohl et al., 2002; Roelofs et al., 2003) and from North America and Asia (at higher altitudes) (Lelieveld et al., 2002). Richards et al. (2013) employed the TOMCAT 3-D chemical transport model to highlight the role of

the South Asian Monsoon outflow in the high ozone concentrations in the middle and upper troposphere over the EM. During summer, biogenic emissions may also influence lower tropospheric ozone via photochemistry (Liakakou et al., 2007) as model estimates suggest that temperature increase may raise biogenic emissions in the region (Im et al., 2011). Finally, the impact of interannual variations in large-scale circulation over the greater region were investigated by Liu et al. (2011). They reported that the interannual variations of ozone transported from Asia and other regions are linked to the position and strength of the

subtropical westerly jet over central Asia.

The circulation over the EM during summer is characterized by a sharp east–west pressure gradient with low pressure over the EMME and high pressure over the western Mediterranean and the Balkans. These large scale synoptic pressure patterns lead to the development of Etesian winds over the Aegean Sea, which are among the most persistent regional wind systems in the world (Carapiperis, 1951; Repapis et al., 1977; Poupkou et al., 2011; Tyrlis et al., 2013; Tyrlis and Lelieveld, 2013;

Anagnostopoulou et al., 2014). The dynamics of the Etesians are tightly interwoven with the large scale dynamics observed over the EM. In fact, the mid-latitude westerlies interact with a zonally asymmetric structure induced by the South Asian Monsoon as a result of westward propagating Rossby waves excited by monsoon convective activity (Rodwell and Hoskins, 1996, 2001; Tyrlis et al., 2013). This, in turn, results in large-scale subsidence over the EM. Subsidence can be further enhanced over southeastern Europe through the diabatic enhancement mechanism described by Rodwell and Hoskins (1996).



Stratosphere-to-troposphere transport (STT) is considered a process of great importance for the EM region, as it influences tropospheric ozone levels during summer (Zanis et al., 2014). Specifically, Zanis et al. (2014) reported that STT processes feed stratospheric ozone into the upper troposphere, and subsequently the ozone-rich air masses are transported to the lower free-tropospheric levels through the characteristic strong summertime EMME subsidence. The main mechanism for STT events

is tropopause folding (Stohl et al., 2003), developed by the ageostrophic flow in the jet stream entrance, which is associated with stratospheric intrusions into the troposphere (Danielsen and Mohnen, 1977). Tropopause folding events mainly occur at mid-latitudes and are characterized by tongues of anomalously high potential vorticity (PV), high ozone and low water vapor mixing ratios (Holton et al., 1995). Recently, Tyrlis et al. (2014) underscored the global hot spot of summertime fold activity between the EM and central Asia, in the vicinity of the subtropical jet, confirming the earlier findings of Sprenger et al. (2003).

Moreover, they reported a striking dynamical link between fold activity over the EMME and the intensity of the South Asian Monsoon on interannual timescales. Convective activity over South Asia was found to regulate upper-level baroclinicity over the EMME and thus the fold occurrence over the region. In summary, the aforementioned studies show that STT events often occur in the EM region and in cases of deep stratospheric intrtusions can reach the lower troposphere (Zanis et al., 2003; Gerasopoulos et al., 2006; Akritidis et al., 2010).

This study aims to assess the contribution of tropopause folds to the summertime pool of high tropospheric ozone concentrations over the EMME. More specifically, the downward transport of ozone-rich air of stratospheric origin is determined with the EMAC model. Another aspect which is explored, is to what extent the interranual variability of near surface ozone over the EM is controlled by the interannual variability of tropopause fold frequency. Section 2 describes the basic features of the EMAC model, elucidates how tropopause folds are identified with the use of a 3-D labeling algorithm and presents the

methodology applied to select the more intense folding events. Section 3 presents the main results regarding the contribution of tropopause folds to tropospheric ozone. In particular, the link between the interannual variability of near surface ozone and that of tropopause fold frequency is investigated. Finally, section 4 summarizes the main conclusions.

## 2    EMAC model description and methodology

### 2.1    EMAC model description

The ECHAM/MESSy Atmospheric Chemistry (EMAC) model is a numerical chemistry and climate simulation system that includes sub-models describing tropospheric and middle atmosphere processes and their interaction with oceans, land and human influences (Jöckel et al., 2016). It uses the second version of the Modular Earth Submodel System (MESSy2) to link multi-institutional computer codes. The core atmospheric model is the 5th generation European Centre Hamburg general circulation model (ECHAM5) (Roeckner et al., 2006). In this work the model results from simulation RC1SD-base-10 of the

ESCiMo project (Jöckel et al., 2016) are used. Detailed information on the model set-up and comparison with observations can be found in Jöckel et al. (2016), while here only basic information on the simulation will be summarized.

The model results were obtained with ECHAM5 version 5.3.02 and MESSy version 2.51, with a T42L90MA-resolution, i.e., with a spherical truncation of T42 (corresponding to a quadratic Gaussian grid of approx. 2.8 by 2.8 degrees in latitude and



longitude) and 90 vertical hybrid pressure levels up to 0.01 hPa. The dynamics of the general circulation model were weakly nudged by Newtonian relaxation towards ERA-Interim reanalysis data (Dee et al., 2011). Model outputs were produced with a 10-hourly temporal resolution.

The simulation RC1SD-base-10 was selected among the ESCiMo simulations because it (i) has been weakly nudged to reproduce "observed" atmospheric dynamics, (ii) has high resolution near the tropopause ($\simeq$ 17 levels between 400 and 100 hPa) for optimal tropopause fold representation and, (iii) is the closest to the one recommended by Jöckel et al. (2016) with enough temporal coverage for climatological study (1979-2013).

Furthermore, besides ozone chemistry, EMAC carries a tracer for stratospheric ozone (denoted by O3s), providing a diagnostic in the investigation of the stratospheric contribution to tropospheric ozone. O3s is set to ozone values in the stratosphere and follows the transport and destruction processes of ozone in the troposphere. When O3s re-enters the stratosphere it is re-initialized at stratospheric values (Roelofs and Lelieveld, 1997).

Figure 1 compares ozonesonde average JJA profiles over Ankara, Turkey (32.86°E: 39.97°N) during the period 1994-2012, obtained from the World Ozone and Ultraviolet Radiation Data Center (WOUDC) (WMO/GAW Ozone Monitoring Community), and the respective EMAC simulated ozone profiles. The corresponding standard deviations are also shown. Overall, the model seems to adequately capture the summertime ozone concentrations throughout the troposphere, although there is a tendency to overestimate them in the middle-upper troposphere. While the modeled ozone variability, as described by the standard deviation, falls within that of the observations, low ozone events tend to be underestimated.

## 2.2 Fold identification algorithm

Tropopause folds are detected in the EMAC model output with the fold identification algorithm by Sprenger et al. (2003), which recently has been improved by Škerlak et al. (2014). A 3-D labeling algorithm is applied to classify the air masses into the following five categories: 1) tropospheric, 2) stratospheric, 3) stratospheric cut-off, 4) tropospheric cut-off, 5) surface-bound PV anomaly. In more detail, the algorithm uses three dimensional fields of potential vorticity, potential temperature and specific humidity and subsequently assigns one of the aforementioned labels in all grid points. A tropopause fold at a grid point is designated where multiple crossings of the dynamical tropopause (2 PVU isosurface) are identified in instantaneous vertical profiles of the label field (for more details see Fig.1 from Škerlak et al. (2015)). Therefore, the 3-D labeling algorithm outputs are 10-hourly binary (1:fold, 0:no fold) data for every grid point and timestep, while the average value of this data for a certain period represents the tropopause fold frequency for the corresponding period. Moreover, for every grid point where a tropopause fold is detected, the upper ($p_U$), middle ($p_M$), and lower ($p_L$) pressure levels of tropopause crossings are determined and subsequently the pressure difference $\Delta p = p_M - p_U$ between the upper and middle tropopause crossings is calculated (for more details see Fig.1 Tyrlis et al. (2014)). The above pressure difference reveals the vertical extent of the tropopause fold (Sprenger et al., 2003; Tyrlis et al., 2014; Škerlak et al., 2015).

Figure 2 presents the summer (JJA) climatology of total folding activity calculated from EMAC simulations. A distinct hot spot of tropopause fold activity is found over the EMME region, as a result of the dynamical interaction between the subtropical jet and the Asian monsoon anticyclone (Tyrlis et al., 2014), with maximum values of total fold frequency up





to 15% over southern Turkey. The above pattern of summertime fold frequency is in line with the results of recent studies (Tyrlis et al., 2014; Škerlak et al., 2015) based on the ERA-Interim reanalysis data. Additionally, a more extensive comparison between EMAC simulated tropopause fold frequency and the results of the aforementioned studies, suggest that both spatial and temporal characteristics of tropopause fold frequencies are well captured by the EMAC modeling system (not shown).

## 2.3 Selection of fold events

In order to study the impact of tropopause folds on summertime tropospheric ozone over the EMME, we selected the most intense summer fold events throughout the period 1979-2013. The influence of tropopause folds on tropospheric ozone over the EMME depends on both the fraction of grid points that exhibit a fold and the vertical extent of the folds. For this purpose, the fold coverage (hereafter FC) within the domain of interest (see marked region in Fig. 2) is calculated, as well as the average vertical extent of the folds, i.e., the average of $\Delta p$ values only for grid points exhibiting a fold. This is done for every summer timestep of the period 1979-2013. Based on FC and $\Delta p$, timesteps are selected when the EMME region is potentially influenced by folds. Figure 3 shows the distribution of FC and $\Delta p$, and the thresholds used to identify intense folds (FC=5.71% and $\Delta p$=60.89 hPa). Thereby, the thresholds are given as the median values of the population of summer timesteps during 1979-2013, but only when at least one grid point with fold is detected inside the domain of interest. Thus, 1866 summer timesteps are selected (see upper-right quartile in Fig. 3), representing 24% of total summer timesteps of the examined period. The intense folds are hereafter called fold events for brevity.

## 3 Results

### 3.1 Tropospheric ozone distribution during folds

Figure 4 (left) shows the composites of ozone concentrations simulated by EMAC that are averaged over the selected fold events. It reveals a pool of high ozone concentrations over the EMME region throughout the free troposphere. More specifically, the highest ozone concentrations in the middle troposphere (Fig. 4a, b and c) are found over the broader EMME region. This high ozone pattern is also evident in the lower troposphere (Fig. 4d) extending geographically to the Persian Gulf. It should be mentioned that this pool of enhanced tropospheric ozone concentrations over the EMME is also seen when all summer timesteps are included (not shown). Hence, it is a persistent feature mainly driven by the summertime circulation and the photochemical regime over the region in agreement with the study by Zanis et al. (2014).

In order to quantify the contribution of stratospheric ozone to the high tropospheric ozone levels during the selected fold events, the ratio of O3s to O3 is investigated. Figure 4 (right) presents at various mid-to-lower tropospheric levels the average of the O3s to O3 ratio during the selected fold events. The percentage (%) of ozone originating from the stratosphere is particularly high over the EMME region, reaching values of about 30-50% in the middle troposphere (Fig. 4e, f and g). A significant contribution of up to 36% is conspicuous even in the lower free troposphere (Fig. 4h).





## 3.2 The impact of tropopause folds on summertime tropospheric ozone

The role of tropopause folds in high tropospheric ozone levels during summer over the EMME is explored next. To this end, anomalies of the average ozone concentrations during fold events are constructed with respect to the concentrations during the rest of summer timesteps (Fig. 5, left). A distinct positive pattern is found in the middle troposphere (Fig. 5a, b and c) mainly

over the EMME region, revealing an increase of ozone up to 7 ppb due to fold activity, which is also clear at 700 hPa (Fig. 5d).

The abovementioned enhancement in ozone levels is due to downward transport of ozone from the stratosphere through the folding process, as can be inferred from the respective anomalies for O3s shown in Figure 5 (right). Similar positive patterns, both quantitatively and spatially, are found for all examined pressure levels (Fig. 5e, f, g and h), supporting the hypothesis that the increase of tropospheric ozone during the selected fold events is mainly attributed to the transport of ozone of stratospheric

origin. Analogous results with the same spatial features but less pronounced positive deviations were obtained by analyzing the anomalies of both O3 and O3s during the selected summer fold events with respect to their summer climatologies (not shown).

## 3.3 Vertical structure and transport

The vertical structure and transport of O3s in the troposphere and the role of tropopause folds is best studied in latitude-pressure and longitude-pressure cross sections of O3s. Figure 6 shows such composite sections for O3s during the selected folds. The

15 main feature depicted in the latitude-pressure cross section 30°E (Fig. 6a) is a remarkable southward and downward intrusion of ozone-rich air towards the lower free tropopshere (approximately down to 800 hPa) within the 15-40°N latitude band. The longitude-pressure cross section of O3s at a latitude of 35°N (Fig. 6b) suggests a similar descending structure of high O3s values in a west-east orientation, over the 20-45°E longitude band. Thus, both vertical cross sections reveal the downward transport pathways of statospheric air masses, resembling the southeastern and downward movement in the vicinity of sharply

sloping isentropes as illustrated in previous stratospheric intrusion case studies over the area (Galani et al., 2003).

To further explore the contribution of tropopause folds to the summertime tropospheric ozone pool over the EMME region, we illustrate the respective vertical cross sections of the anomalies of O3s during the selected fold events with respect to the rest of summer timesteps (Fig. 6c and d). The anomalies of EMAC simulated O3s latitude-pressure cross sections at a longitude of 30°E (Fig. 6c) indicate a clear increase of O3s throughout the free tropospere during the selected fold events, with values

of up to 10 ppb in the upper troposphere, which decreases towards lower tropospheric levels following the sloping isentropes. The same picture emerges from the longitude-pressure cross section at a latitude of 35°N (Fig. 6d).

Figure 7a depicts the vertical profiles of the anomalies of O3s during the selected fold events with respect to the rest of summer timesteps for 5 grid points which are located in a north-west directed axis over the EMME region (Fig. 7b). A clear positive anomaly of up to 15 ppb is found in the upper troposphere over the Aegean Sea extending down to roughly 600 hPa

(grid point 1). Further southeastwards, the peak of the positive ozone anomaly gradually occurs lower in the free tropopshere. For example, at grid point 2 (near to Crete) the maximum anomaly of around 11 ppb at 350 hPa extends down to 700 hPa, while grid point 3 indicates a peak anomaly of around 8 ppb at 450 hPa extending down to 800 hPa. Further downstream, grid points 4 and 5 reveal peak anomalies of about 8 and 6 ppb at 550 hPa and 600 hPa respectively, extending down to 900 hPa




over grid point 5. This is in agreement with the studies by Zanis et al. (2014) and Tyrlis et al. (2014) that provided evidence of a southeastward migration of the maxima of high-PV and ozone anomalies closer to the surface. This is due to the fact that ozone-rich and high PV air masses follow the sharply sloping isentropes, in an almost adiabatic fashion, as they spread from the Balkans toward the Levantine region. As also noted by Tyrlis et al. (2014) the deeper but rarer folds are identified over the

Levantine region, which is in agreement with the fact that ozone anomalies at lower level are located over this region.

### 3.4    Interannual variability

To investigate the possible impacts of tropopause folds on both tropospheric ozone and near surface ozone over the EM region, we examined the year-to-year relation between tropopause fold frequency, EMAC simulated free tropospheric ozone and near surface ozone observations. For this purpose we consider near surface ozone data from the baseline maritime station at

Finokalia-Crete (GR02, 25.67°E: 35.32°N, see Fig. 9 for location) from the European Monitoring and Evaluation Programme (EMEP) network for the time period 1998-2013. Figure 6b suggests that in the EMAC simulations the contribution of the stratospheric reservoir does not reach the surface in the vicinity of Crete during the selected fold events. However, this could be related to other model processes that partly mask the contribution of downward transport to the near surface level, such as overestimation of photochemical ozone production related to emission inventories and the coarse horizontal resolution, as well

as issues related to the accurate representation of processes that determine the entrainment from the lower free troposphere into the atmospheric boundary layer (Zanis et al., 2014).

Hereafter, the Pearson correlation coefficient is used to quantify the relationship between EMAC ozone, fold frequency and surface ozone observations. Its significance at the 95% confidence level is assessed based on t-test statistics. First, the interannual variability of the mean July-August tropopause fold frequency over a southern Balkan region (hereafter SB, 20-

27°E, 37-44°N) is found to be positively correlated at the 95% significance level with the mean July- August EMAC simulated O3 and O3s (r=0.69 and r=0.65 respectively) in the lower free troposphere (700 hPa) over the EM (20-30°E, 30-40°N) (Fig. 8). Similarly, the mean July-August ozone concentration measurements at Finokalia are also positively correlated at the 95% significance level with the mean July-August tropopause fold frequency over SB (r=0.64) (Fig. 8). Moreover, the observed ozone values at Finokalia show a positive correlation at the 95% significance level with both O3 and O3s values at 700 hPa

over the EM, with values of r=0.6 and r=0.52 respectively (Fig. 8). All in all, the correlations indicate a link of the observed near surface ozone at Finokalia with both tropopause fold frequency and ozone of stratospheric origin at the lower free troposphere.

The findings above are further supported when we consider the spatial distribution of the correlation coefficient between the mean July-August values of the observed ozone at Finokalia and the tropopause fold frequency at each grid point (Fig. 9). Indeed, positive correlations at the 95% significance level can be found over northern Greece and the central Mediterranean.

The sharp change in the sign of the correlation just to the north of Crete could be interpreted by the morphology of folds and the associated STE. Typically a fold advances from the northwest towards the southeast and the intrusion of high PV and ozone air that develops also moves downwards and southwards. The fold becomes mature and sooner or later it "breaks" and fragments of high PV can disperse downward. If such patches of high-PV and ozone-rich air survive subsidence towards the surface



near Crete, this can only be associated with folding occuring further upstream over the Balkans. This is due the background northwesterly flow over the region and the morphology of intrusions described in Figure 6.

In order to investigate in more detail the links between ozone of stratospheric origin with both tropopause fold frequency over SB and near surface ozone at Finokalia, we constructed the corresponding latitude-pressure cross-sections at 25°E (parallel crossing the island of Crete and the Aegean Sea) of the correlation coefficient (Fig. 10a and b respectively) between the mean July-August interannual timeseries of the period 1998-2013. Figure 10a indicates a significant positive correlation of the mean July-August tropopause fold frequency over SB with the mean July-August EMAC simulated O3s in the lower free troposphere over the 25-40°N latitude band. Similarly, significant positive correlations between the mean July-August values of observed ozone at Finokalia and mean July-August EMAC simulated O3s are found in the middle/lower troposphere. The vertical structure of the positive correlations in both latitude-pressure cross sections in Figure 10 resembles the structure of the latitude-pressure cross-section 25°E of O3s (not shown), thus indicating the dynamical nature of the link between both observed near surface ozone at Finokalia and tropopause fold frequency over SB with EMAC simulated ozone of stratospheric origin. Nevertheless, no significant correlation is found between the observed and EMAC simulated near surface ozone values at Finokalia, which could be related to overestimated photochemical ozone production in the boundary layer by the model.

Based on the results so far, we infer that the interannual variability of near surface ozone over the EM is partly governed by the interrannual variabilities of tropospheric ozone of stratospheric origin and tropopause folds. This hypothesis is further supported when the trends of both tropopause fold frequency and EMAC simulated O3s are considered (Fig. 11). The trends during summer over the period 1979-2013 were calculated by implementing linear regression analysis on mean summer values, while the statistical significance of the trends is assessed using the Mann-Kendall test (Press et al., 1992) at the 95% confidence level. During summer, an elongated zone of positive fold frequency trends is detected across EM, Turkey and Caspian Sea with values of up to 0.3 %/year (Fig. 11a). Similar positive shallow fold (with the depth of the fold ranging between 50 hPa and 200 hPa) frequency trends during July-August of the period 1979-2012 have been reported by Tyrlis et al. (2014) using the ERA-Interim reanalysis data. The spatial distribution of O3s trends during summer at 400 hPa (Fig. 11b) indicates a distinct area of positive trends mostly over EMME, while towards lower tropospheric levels at 500 hPa, 600 hPa and 700 hPa the positive trend signal remains but attenuates (Fig. 11c, d and e). This increase in summertime EMAC simulated O3s may be associated with the aforementioned positive trends in fold frequency, as no significant trend in ozone total column (not shown) was found during summer over the EMME region. Our results are in line with Ordóñez et al. (2007) who pointed out that both the effects of stratospheric ozone and STT changes need to be represented accurately in models in order to describe the evolution of the background tropospheric ozone.

## 4   Conclusions

We investigated the role of tropopause folds in the formation of the summertime ozone pool over the EMME with the aid of simulations covering the period 1979-2013 by the atmospheric chemistry-climate model EMAC. Tropopause fold events in



EMAC simulations were identified with the aid of the updated 3-D labeling algoritm (Škerlak et al., 2014) initially developed by Sprenger et al. (2003). The most noteworthy results in this study can be summarized as follows:

- A summertime hot spot of tropopause fold occurence is identified in EMAC simulations over the EMME region, which agrees with the results of previous studies (Sprenger et al., 2003; Tyrlis et al., 2014; Škerlak et al., 2015), indicating that the fold activity during summer over the region is well captured by the EMAC modeling system.

- A distinct pool of high ozone concentrations is found in the middle troposphere over the EMME during the selected fold events. Moreover, the EMAC simulated O3/O3s ratio, averaged over all the selected fold events, implies a significant contribution of stratospheric ozone to the high tropospheric ozone mixing ratios over the EMME even in the lower free troposphere.

- The pool of high tropospheric ozone over the EMME is a clear and permanent feature during summer, as was also pointed out by Zanis et al. (2014). The anomalies of EMAC simulated O3 and O3s during the selected fold events relative to the remainder of summer timesteps reveal the key role of tropopause folds in stratospheric ozone-rich air mass transports into the troposphere. A considerable enhancement for both O3 and O3s is identified in the middle troposphere over the EMME extending down to the lower free troposphere, as a result of the fold activity over the region.

- In agreement with Zanis et al. (2014), the location of the ozone maximum during the selected fold events shifts towards the southeast with decreasing altitude. The contribution of tropopause folds in middle/lower tropospheric ozone seems to be most significant over the south-eastern Mediterranean, as a result of the vertical downward transport of stratospheric ozone and the prevailing north-westerly flow in the middle/lower free troposphere during summer.

- A year-to-year analysis indicates a relation between the observed surface ozone at Finokalia with both tropopause fold frequency and tropospheric ozone of stratospheric origin in the middle/lower free troposphere over the EM, as the corresponding correlation coefficients were found positive and statistically significant. Hence, tropopause folds over the southern Balkans and O3s at 700 hPa over the EM explain 41% and 27%, respectively, of the interannual variability of the mean July-August values of surface ozone at Finokalia for the time period 1998-2013.

- Finally, the upward trend of EMAC simulated O3s in the upper/middle troposphere during summer over the EMME appears to be partly controlled by the corresponding trend in tropopause fold frequency. Taken together, this suggests that tropopause folds and tropospheric ozone with stratospheric origin have a greater impact on summertime near surface ozone over the EM than previously thought, and contribute significantly to near surface ozone interannual variability.



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





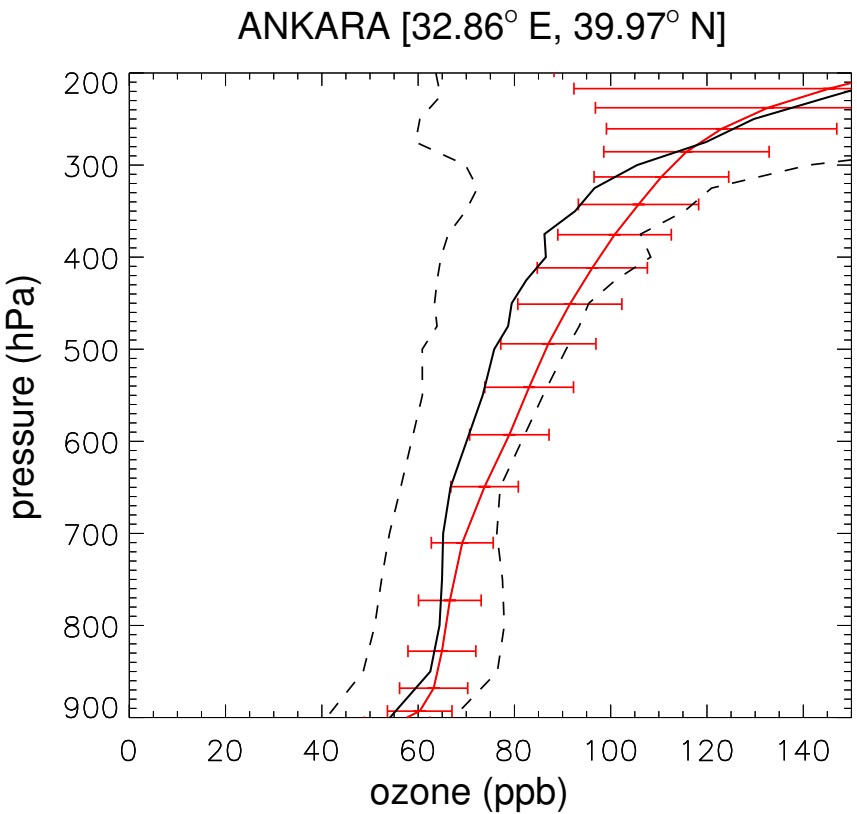

**Figure 1.** Vertical profiles of JJA ozone (mixing ratios) over Ankara (STN348) for the period 1994-2012. Solid black line represents observations and red line refers to EMAC simulated ozone. The dashed black lines show the observed standard deviations and the red bars show the model standard deviations.





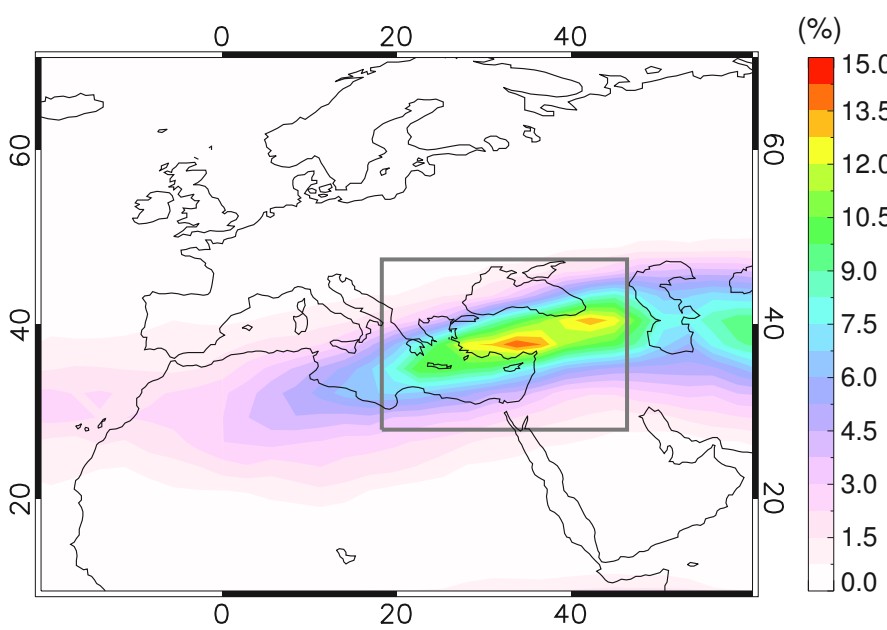

**Figure 2.** Mean tropopause fold frequencies (%) during summer for the period 1979-2013 from EMAC simulations. The box indicates the domain of interest (see Section 2.3).



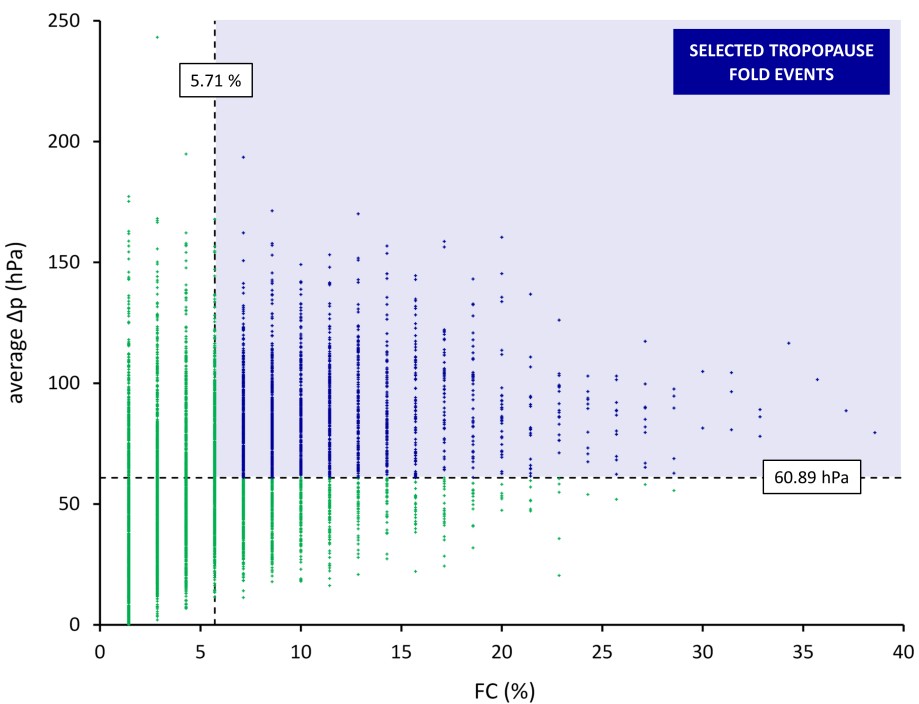

**Figure 3.** Scatterplot distributions of FC and average Δp for the summer timesteps of the period 1979-2013 over the domain of interest (box in Fig. 2).





**Figure 4.** Spatial distribution of EMAC simulated ozone (ppb) and stratospheric ozone contribution (%) averaged over the selected fold events of the period 1979-2013 at 400 hPa (a, e), 500 hPa (b, f), 600 hPa (c, g) and 700 hPa (d, h).





**Figure 5.** Anomalies of EMAC simulated ozone (left) and stratospheric ozone tracer (right) during the selected fold events from the remainder summer timesteps at 400 hPa (a, e), 500 hPa (b, f), 600 hPa (c, g) and 700 hPa (d, h).





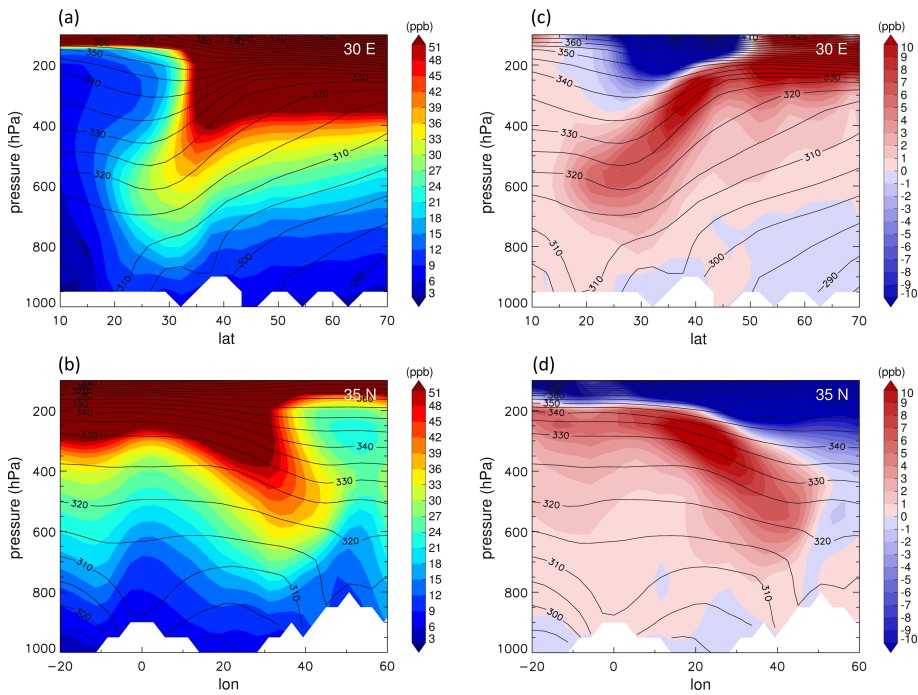

**Figure 6.** Latitude-pressure cross sections at 30°E of (a) O3s during the selected fold events and (c) the anomalies from the remainder summer timesteps. Longitude-pressure cross sections at 35°N of (b) O3s during the selected fold events and (d) the anomalies from the rest of summer timesteps. Black contours denote potential temperature (K) during the selected fold events.





**Figure 7.** Vertical profiles of the anomalies of EMAC simulated O3s during the selected fold events with respect to the rest of summer timesteps (a) for 5 grid points in a north-west direction over the EMME region (b).





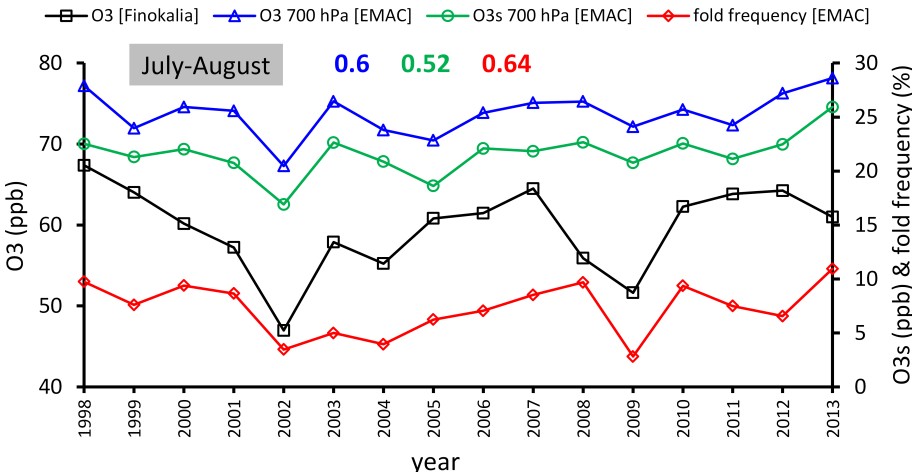

**Figure 8.** Time series over the period 1998-2013 of July-August average values for observed near-surface ozone at Finokalia Crete (black squares), EMAC simulated ozone at 700 hPa over EM (blue triangles, 20-30°E, 30-40°N), EMAC simulated O3s at 700 hPa over EM (green circles, 20-30°E, 30-40°N) and tropopause fold frequency over SB (red diamonds, 20-27°E, 37-44°N). The colored numbers above the lines are the correlation coefficients between July-August average values of near surface ozone at Finokalia and EMAC simulated O3 at 700 hPa over EM (blue); EMAC simualted O3s at 700 hPa over EM (green); tropopause fold frequency over SB (red) respectively.





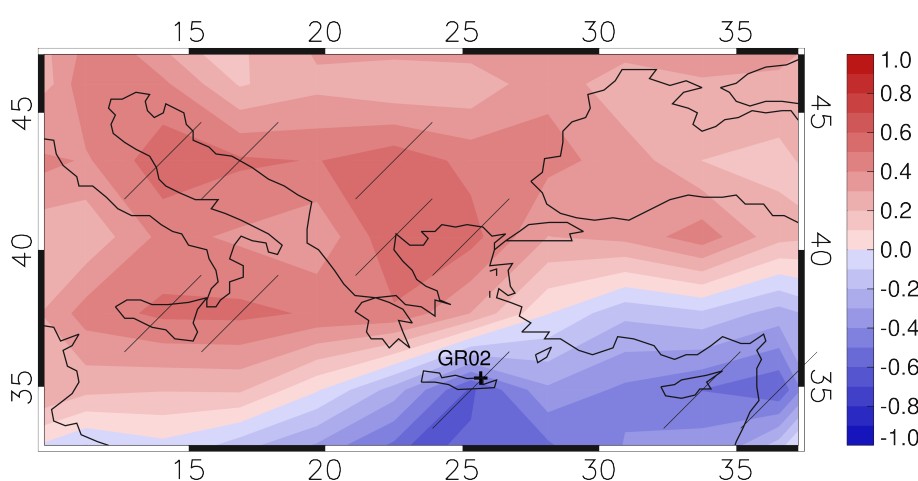

**Figure 9.** Spatial distribution of the correlation coefficient between July-August average values of near surface ozone at Finokalia and July-August average values of tropopause fold frequency at each grid point over the period 1998-2013. Areas featuring correlation coefficients that are statistically significant at the 95% confidence level are hatched.



**Figure 10.** Latitude-pressure cross-sections at 25°E of the correlation coefficient between the mean July-August interannual timeseries (1998-2013) of (a) tropopause fold frequency over SB and EMAC simulated O3s and (b) near surface ozone at Finokalia and EMAC simulated O3s. Areas featuring correlation coefficients that are statistically significant at the 95% confidence level are hatched.



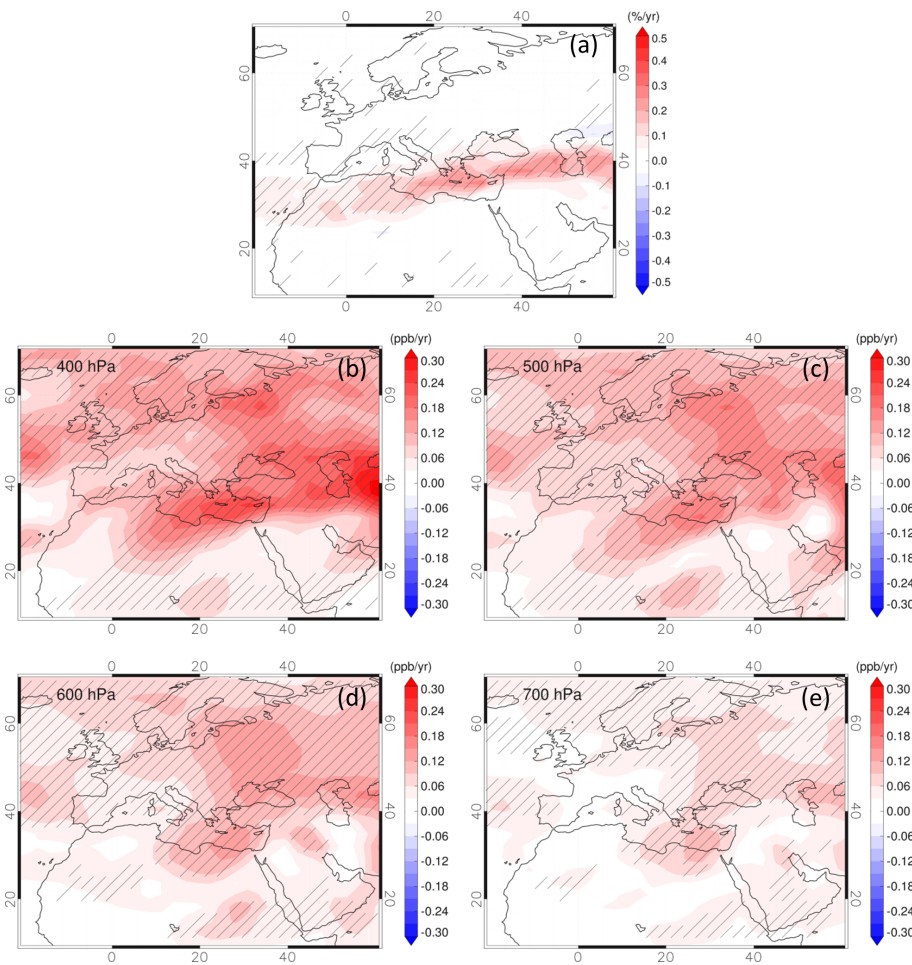

**Figure 11.** Trends of (a) JJA tropopause fold frequency (%/yr) and JJA EMAC simulated O3s (ppb/yr) at (b) 400 hPa, (c) 500 hPa, (d) 600 hPa and (e) 700 hPa during the period 1979 -2013. Areas featuring trends that are statistically significant at the 95% confidence level are hatched.