# Peer review of "On the role of tropopause folds in summertime tropospheric ozone over the eastern Mediterranean and the Middle East"

_Atmospheric Chemistry and Physics, 2016_

## Referee Comment (RC1) · Anonymous Referee #1 · 8 Aug 2016

This paper investigates the role of tropopause folds in controlling high tropospheric ozone concentrations that are consistently observed in the eastern Mediterranean and the Middle East (EMME) region. The ECAM model is used to assess the frequency of strong tropopause fold events, the percentage of ozone in this region that has been transported from the stratosphere, and the interannual variability of tropopause folds. The analysis is well presented and, as the global community continues to work to set stringent yet attainable air quality standards, the topic should be of great interest to ACP readers. I believe that there are several issues, described below, that are deserving of more discussion and analysis, but recommend publication following these minor revisions.

[Figure]

Specific comments: P3, L33 – The T42 resolution (∼2.8 degree) is fairly coarse for resolving these events. Finer scale models can show substantial differences in the amount of stratospheric ozone transported to the mid- and lower troposphere (e.g. see Lin et al., 2015 supplementary material). Though this resolution may be necessary for the longer timescale simulation presented in this work, the potential impact should be noted and discussed.

P4, L8-11 – Zhang et al. (2014) showed that diagnosing the stratospheric influence over the United States was strongly dependent on how stratospheric ozone was defined (e.g. ozone produced in the stratosphere or ozone transported from the above the tropopause). It would be good to mention and discuss the implications of this assumption.

P6, L5 – 7 ppb is the mean enhancement for the tropopause fold composite, but what is the range? Since many people may be interested in this work from an air quality perspective, a discussion not just of the mean case but the extremes would be of great interest.

Lin, M., A.M. Fiore, L.W. Horowitz, A.O. Langford, S. J. Oltmans, D. Tarasick, H.E. Reider (2015), Climate variability modulates western US ozone air quality in spring via deep stratospheric intrusions, Nature Communications, 6, 7105, doi:10.1038/ncomms8105.

Zhang, L., D. J. Jacob, X. Yue, N. V. Downey, D. A. Wood, and D. Blewitt (2014), Sources contributing to background surface ozone in the US intermountain West, Atmos. Chem. Phys., 14, 5295-5309.

---

## Referee Comment (RC2) · Anonymous Referee #2 · 7 Sep 2016

The authors have used the ECHAM5/MESSy Atmospheric Chemistry model (EMAC) to quantify the influence of ozone transported from the stratosphere (through tropopause folds) on tropospheric ozone abundances over the eastern Mediterranean and the Middle East. This is a region that is a crossroads for transport of pollution from Europe, Asia, and North America and where there is persistent high summertime abundances of tropospheric ozone. In recent years there has been an increasing number of studies focusing on ozone in this region. This manuscript is a valuable addition to this growing body of literature. It helps establish the importance of tropopause folding events as a key mechanism driving the summertime buildup of ozone over the Mediterranean and the Middle East. I would therefore recommend the manuscript for publication after the

authors have addressed my comments below.

Main Comments

1) The model resolution is 2.8 x 2.8 degrees, which is coarse compared to studies such as Lin et al. (JGR, doi:10.1029/2012JD018151, 2012), who used a model with a resolution of 0.5 x 0.5 degrees to study stratospheric intrusions. Although the EMAC model is nudged toward ECMWF ERA-Interim data, it would be helpful to see how the model compares to the ERA-Interim fields. On page 5, lines 2-4, it states that "a more extensive comparison . . . suggest that both spatial and temporal characteristics of tropopause fold frequencies are well captured by the EMAC modeling system (not shown)." However, given the coarse model resolution used here, I believe that it is important to show the results of these comparisons to establish the fidelity of the model in capturing tropopause folds.

2) It is unclear how the anomalies that are shown in Figures 5-7 are calculated. More information than what is given in line 3 on page 6 would be helpful and would make it easier to interpret the results that are presented.

Minor comments

1) Page 1, lines 13-16: This is a long sentence. Please try breaking it into two sentences, separately addressing the long-range transport and radiative effects.

2) Page 1, lines 16-18: The sentence starting with "Tropospheric ozone originates. . ." is long and difficult to read. Also, What about methane? It is not a volatile organic compound, but it is an important precursor of tropospheric ozone.

3) Page 2, line 2: Li et al. (GRL. Vol. 28, 3235-3238, 2001), which first highlighted the summertime ozone buildup over the Middle East, should be referenced here.

4) Page 4, line 6: "Optimal" in what sense? How was it determined that the vertical resolution should be for optimal tropopause fold representation?

5) Page 6, line 5: At what level is this 7 ppb increase found? What is the maximum increase in ozone at 700 hPa? The values at 700 hPa seem to be much less than 7 ppb.

6) Figure 6: What do the negative values in the stratosphere mean in panels c) and d)?

7) Figure 6: The anomalies seem to be descending across the potential temperature surfaces, indicating that the downward transport is not purely isentropic. In this region descent is also associated with radiative cooling. How consistent is the rate of cross-isentropic transport with the cooling rates in this region and the timescale for downward transport in these folding events?

---

## Author Response (AR1)

**Response to the Referees**

*"On the role of tropopause folds in summertime tropospheric ozone over the eastern Mediterranean and the Middle East"*

by Dimitris Akritidis et al.

Dear Editor,

At first we would like to thank the referees for their helpful comments and their time devoted on reviewing our manuscript.

On the following pages, we present our point by point response (teal color) to all the comments raised by the reviewers (black color), along with the corresponding changes made in the manuscript. A version of the revised manuscript with our changes highlighted (blue and red) is also included. Please also note that two new figures are added as supplementary material in support of the revised manuscript.

Sincerely,
Dimitris Akritidis (on behalf of all co-authors)

Anonymous Referee #1

We would like to thank Reviewer #1 for his/her time devoted and the constructive and helpful comments.

This paper investigates the role of tropopause folds in controlling high tropospheric ozone concentrations that are consistently observed in the eastern Mediterranean and the Middle East (EMME) region. The ECAM model is used to assess the frequency of strong tropopause fold events, the percentage of ozone in this region that has been transported from the stratosphere, and the interannual variability of tropopause folds. The analysis is well presented and, as the global community continues to work to set stringent yet attainable air quality standards, the topic should be of great interest to ACP readers. I believe that there are several issues, described below, that are deserving of more discussion and analysis, but recommend publication following these minor revisions.

We thank the Reviewer for the comments, to which we will respond point by point.

Specific comments:

P3, L33 – The T42 resolution (~2.8 degree) is fairly coarse for resolving these events. Finer scale models can show substantial differences in the amount of stratospheric ozone transported to the mid- and lower troposphere (e.g. see Lin et al., 2015 supplementary material). Though this resolution may be necessary for the longer timescale simulation presented in this work, the potential impact should be noted and discussed.

We agree with the Reviewer, that the potential implications of a finer resolution should be mentioned and further discussed in the manuscript. The following paragraph was added in the revised manuscript (Page 3, Lines 20-28): *"Model sensitivity studies suggest that relatively high horizontal resolution might be beneficial for the representation of tropopause fold events and the associated intrusion of stratospheric ozone into the troposphere (Kentarchos et al., 2000). Lin et al. (2012) showed that a global high-resolution (50 km x 50 km) chemistry-climate model (GFDL AM3) captures the observed layered features and sharp ozone gradients of deep stratospheric intrusions. Moreover, Lin et al. (2015) carrying out sensitivity studies with the GFDL AM3 model, pointed out that using a finer horizontal resolution of 50 km x 50 km revealed an improvement in the reproduction of the day-to-day variability in the upper troposphere, as tropopause fold filamentary structures are better resolved in the finer model resolution simulations. Nevertheless, they suggested that the multidecadal hindcast simulations with the coarser resolution of 200 km x 200 km were also found suitable for quantifying the regional-scale interannual variability of stratospheric influence on*

*lower tropospheric ozone over the western US."* Please also note, that according to a comment from Reviewer #2 regarding the coarse resolution of EMAC and its respective capability to reproduce the features of tropopause folds frequency, we included Figure S1 (Supplementary Material) which reveals a good agreement between the EMAC-simulated tropopause fold activity and the results of the study by Tyrlis et al. (2014) which was based on the ERA-Interim reanalysis data (see discussion at the revised manuscript at Page 5, Lines 15-21).

P4, L8-11 – Zhang et al. (2014) showed that diagnosing the stratospheric influence over the United States was strongly dependent on how stratospheric ozone was defined (e.g. ozone produced in the stratosphere or ozone transported from the above the tropopause). It would be good to mention and discuss the implications of this assumption.

We agree with the Reviewer that different definitions of stratospheric ozone tracer are likely to have different impacts on stratospheric ozone contribution to tropospheric ozone. To this end, and following the reviewer's suggestion, the following discussion was included in the revised manuscript (Page 7, Lines 5-13):
*"It should be mentioned that different definitions of stratospheric ozone tracer may have implications for the estimated stratospheric contribution, as has been pointed out by Zhang et al. (2014). More specifically, a doubling of the diagnosed stratospheric ozone influence was found in GEOS-Chem simulations when the produced ozone in the troposphere, which was temporarily transported above the tropopause, was also considered as stratospheric (Lin et al. (2012) approach). The amount of ozone that is recirculated across the tropopause in the model depends on the vertical resolution. Thus, following the approach by Lin et al. (2012) is likely to yield an upper limit to the stratospheric contribution to tropospheric ozone in our results. Nevertheless, because the presently used middle atmosphere version of the EMAC model has relatively high resolution in the upper troposphere and lower stratosphere (about 500m), and because we initialize O3s well above the tropopause (100 hPa), we expect this effect to be small."*

P6, L5 – 7 ppb is the mean enhancement for the tropopause fold composite, but what is the range? Since many people may be interested in this work from an air quality perspective, a discussion not just of the mean case but the extremes would be of great interest.

We thank the Reviewer for the suggestion. As it is mentioned in the manuscript, 7 ppb is the maximum O3 enhancement in the middle troposphere (400, 500 and 600 hPa) due to fold activity, resulting from the differences between the average O3 during fold events and the average O3 during the remainder summer timesteps. To

present the range of tropopause folds impact on tropospheric ozone, we present in Figure S2 (Supplementary Material) the box-whisker plots of O3 concentrations (averaged over the regions where the positive anomalies in Fig. 5 are found) during the fold events, along with the respective average O3 concentrations during the remainder summer timesteps. According to this, the following sentence was added in revised manuscript (Pages 6, Lines 27-29): *"During extreme events (above the 95th percentile of O3 concentrations during fold events), the range of O3 enhancement is found to be 19-33, 16-31, 17-24 and 11-19 ppb at 400, 500, 600 and 700 hPa respectively (Fig. S2 in the Supplementary Material)."*

[Figure]

**Figure S2.** Box-whisker plots of O3 concentrations during fold events at 400, 500, 600 and 700 hPa. The middle line in the box shows the median O3 concentrations, the box represents the range between the 25th and 75th percentiles, the top/bottom whiskers indicate the max/min concentrations and the filled black square shows the 95th percentile. The filled blue triangle represents the average O3 concentrations during the remainder summer timesteps. O3 concentrations are calculated as the spatial average of the regions (orange table) where the positive anomalies in Fig. 5 are mainly found.

Anonymous Referee #2

We would like to thank Reviewer #2 for his/her time devoted and the constructive and helpful comments.

The authors have used the ECHAM5/MESSy Atmospheric Chemistry model (EMAC) to quantify the influence of ozone transported from the stratosphere (through tropopause folds) on tropospheric ozone abundances over the eastern Mediterranean and the Middle East. This is a region that is a crossroads for transport of pollution from Europe, Asia, and North America and where there is persistent high summertime abundances of tropospheric ozone. In recent years there has been an increasing number of studies focusing on ozone in this region. This manuscript is a valuable addition to this growing body of literature. It helps establish the importance of tropopause folding events as a key mechanism driving the summertime buildup of ozone over the Mediterranean and the Middle East. I would therefore recommend the manuscript for publication after the authors have addressed my comments below.

We thank the Reviewer for the comments, to which we will respond point by point.

Main Comments

1) The model resolution is 2.8 x 2.8 degrees, which is coarse compared to studies such as Lin et al. (JGR, doi:10.1029/2012JD018151, 2012), who used a model with a resolution of 0.5 x 0.5 degrees to study stratospheric intrusions. Although the EMAC model is nudged toward ECMWF ERA-Interim data, it would be helpful to see how the model compares to the ERA-Interim fields. On page 5, lines 2-4, it states that "a more extensive comparison … suggest that both spatial and temporal characteristics of tropopause fold frequencies are well captured by the EMAC modeling system (not shown)." However, given the coarse model resolution used here, I believe that it is important to show the results of these comparisons to establish the fidelity of the model in capturing tropopause folds.

We agree with the Reviewer. As the model horizontal resolution was also mentioned by Reviewer #1, we have included a new paragraph in the revised manuscript (Page 3, Lines 20-28), where we further discuss the benefits of a finer horizontal resolution. Following the Reviewer's suggestion, we have included as Supplementary Material (Figure S1) the EMAC monthly mean climatology of shallow ($50 \leq \Delta p < 200$ hPa), medium ($200 \leq \Delta p < 350$ hPa) and deep ($\Delta p \geq 350$

[Figure]

**Figure S1.** Monthly mean climatology (1979-2012) of shallow (50 ≤ Δp < 200 hPa), medium (200 ≤ Δp < 350 hPa) and deep (Δp ≥ 350 hPa) fold frequency during March, May, June, July and September, for intercomparison with Fig. 2 of Tyrlis et al. (2014).

hPa) fold frequency during March, May, June, July and September, for a straightforward intercomparison with the results of the study by Tyrlis et al. (2014). The selection of the months by Tyrlis et al. (2014) is based on the seasonal evolution of both tropopause folds and south Asian Monsoon activities. The results of this comparison are further discussed in the revised manuscript and the respective paragraph has been modified as follows (Page 5, Lines 15-26): *"To evaluate the ability of the EMAC model to capture tropopause fold activity, we compare results with the findings of Tyrlis et al. (2014), based on the ERA-Interim reanalysis data. The monthly mean climatology (1979-2012) of shallow (50 ≤ Δp < 200 hPa), medium (200 ≤ Δp < 350 hPa) and deep (Δp ≥ 350 hPa) fold frequency during several months is depicted in Figure S1 (Supplementary Material), for intercomparison with Figure 2 of Tyrlis et al. (2014). Both temporal and spatial patterns of EMAC-simulated shallow (more frequent) and medium fold frequencies are found to be in good agreement with the ERA-Interim reanalysis data. The very*

*rare occurence of deep folds in ERA-Interim data (with a peak frequency of about 0.1%) is not reproduced by EMAC, probably due to its coarser horizontal resolution. Figure 2 presents the summer (JJA) climatology of total folding activity calculated from EMAC simulations. A distinct hot spot of tropopause fold activity is found over the EMME region, as a result of the dynamical interaction between the subtropical jet and the Asian monsoon anticyclone (Tyrlis et al., 2014), with maximum values of total fold frequency up to 15% over southern Turkey. The above pattern of summertime fold frequency is in line with the results of recent studies (Tyrlis et al., 2014; Škerlak et al., 2015) based on the ERA-Interim reanalysis data.".*

2) It is unclear how the anomalies that are shown in Figures 5-7 are calculated. More information than what is given in line 3 on page 6 would be helpful and would make it easier to interpret the results that are presented.

We agree with the Reviewer's suggestion to provide more information on how the anomalies are calculated. The following sentence was included in the revised manuscript (Page 6, Lines 21-23): *"The anomalies of O3 and O3s presented hereafter, are calculated as the differences between the average concentrations during fold events (average over 1866 timesteps) and the average concentrations during the remainder summer timesteps (average over 5864 timesteps)."*

Minor comments

1) Page 1, lines 13-16: This is a long sentence. Please try breaking it into two sentences, separately addressing the long-range transport and radiative effects.

Done. We have split the respective sentence into two as follows (Page 1, Lines 13-16): *"Compared to ozone near the surface, ozone in the free troposphere can be transported over greater distances due to its relatively longer lifetime and the higher wind velocities. Moreover, owing to its high radiative forcing efficiency in the upper troposphere ozone concentration changes have proportionally greater impact on climate compared to the lower troposphere (Lacis et al., 1990)."*

2) Page 1, lines 16-18: The sentence starting with "Tropospheric ozone originates…" is long and difficult to read. Also, What about methane? It is not a volatile organic compound, but it is an important precursor of tropospheric ozone.

We agree with the Reviewer that methane is also an important ozone precursor, which reaction with hydroxyl radical initiates a sequence of reactions that result in ozone production. To this end, methane was also included as an ozone precursor in the revised manuscript. Following the Reviewer's suggestion we have modified the respective sentence as follows (Page 1, Lines 16-19): *"The main sources of ozone in the troposphere are (i) photochemical production through a sequence of reactions from its precursors (nitrogen oxide, volatile organic compounds, carbon*

*monoxide and methane) (Crutzen, 1974) and, (ii) downward transport from the stratosphere (Danielsen, 1968)."*

3) Page 2, line 2: Li et al. (GRL. Vol. 28, 3235-3238, 2001), which first highlighted the summertime ozone buildup over the Middle East, should be referenced here.
Done (Page 2, Line 2). We thank the Reviewer for noticing this out.

4) Page 4, line 6: "Optimal" in what sense? How was it determined that the vertical resolution should be for optimal tropopause fold representation?
What the authors meant is that the high vertical resolution of the model near the tropopause contributes to a more realistic representation of the tropopause fold process. Therefore, we have replaced the word *"optimal"* with the word *"realistic"* (Page 4, Line 19).

5) Page 6, line 5: At what level is this 7 ppb increase found? What is the maximum increase in ozone at 700 hPa? The values at 700 hPa seem to be much less than 7 ppb.
We thank the Reviewer for this comment. Indeed the values at 700 hPa are not up to 7 ppb as inadvertently is stated in the manuscript. The increase of ozone due to fold activity is up to 7 ppb at 400, 500 and 600 hPa, and up to 4 ppb at 700 hPa. The respective discussion has been modified accordingly in the revised manuscript as follows (Page 6, Lines 25-27): *"A distinct positive pattern is found in the middle troposphere (Fig. 5a, b and c) mainly over the EMME region, revealing an increase of ozone up to 7 ppb due to fold activity. An increase of ozone is also clear at 700 hPa (Fig. 5d) with mixing ratios of up to 4 ppb."*

6) Figure 6: What do the negative values in the stratosphere mean in panels c) and d)?
The positive O3s anomalies in the troposphere and the negative O3s anomalies aloft are simply due to mass redistribution in the vertical, taking into consideration the mass balance.

7) Figure 6: The anomalies seem to be descending across the potential temperature surfaces, indicating that the downward transport is not purely isentropic. In this region descent is also associated with radiative cooling. How consistent is the rate of cross isentropic transport with the cooling rates in this region and the timescale for downward transport in these folding events?
Despite the isentropic framework (in fact near isentropic) of stratospheric intrusions that result on the formation of filamentary structures within the troposphere, the lifetime and evolution of these filaments are influenced by non-conservative processes such as radiative cooling and heating, turbulence (e.g. generated by

wind shear, convection, breaking gravity waves and radiation) and molecular diffusion, which cascade the filaments gradually to smaller scales down to the molecular level (Forster and Wirth, 2000; Stohl et al., 2003).

The point raised by the Reviewer for the role of radiative cooling in the evolution of the intruded filamentary structures is an interesting point, but at the same time a complicated issue, which cannot be readily unraveled in our case from Figure 6 which shows a mean situation of a number of different selected cases characterized as fold events. This issue could be more readily addressed through idealized model experiments or specific case studies, where we could isolate the effect of radiative cooling from other important diabatic processes, such as turbulent mixing. For example, Forster and Wirth (2000) addressed the issue of radiative decay of stratospheric filaments in the troposphere in idealized model experiments for different filaments (thick or thin) and assuming solely a PV anomaly or a PV anomaly associated with an anomaly in ozone and water vapour, without taking into account turbulent mixing.

A quantitative answer on Reviewer's question is beyond the scope of this manuscript, but we address this point and the relevant limitations within the manuscript as follows (Page 3, Lines 8-12): *"Subsequent to transport into the troposphere, air with stratospheric origin is quasi-adiabatically stirred by large-scale cyclonic and anticyclonic disturbances, which may lead to the formation of elongated streamers or isolated coherent structures. These can further dissipate and cascade down to smaller scales by non-conservative processes (such as radiative cooling/heating and turbulence), thus leading to irreversible mixing with the surrounding air (Shapiro, 1980; Appenzeller and Davies,1992; Forster and Wirth, 2000)."*

**Please also note a few minor changes in the manuscript:**

- Page 2, Line 21 "also" is removed.
- Page 2, Line 22 *"temperature increase may raise"* is replaced with *"climate warming may intensify"*.
- Page 3, Line 2 *"Specifically,"* is removed.
- Page 3, Line 31 *"which is explored"* is replaced with *"explored here"*.
- Page 4, Line 20 *"enough"* is replaced with *"sufficient"*.
- Page 4 Lines 24-25 the phrase *", however, since it is initialized above 100 hPa, only a very small fraction is recirculated by multiple crossings of the tropopause"* is added.
- Page 10, Line 5 *"modeling system"* is replaced with *"model"*.

**The following references have been added in the revised manuscript:**

Appenzeller, C. and Davies, H.: Structure of stratospheric intrusions into the troposphere, 1992.

Forster, C. and Wirth, V.: Radiative decay of idealized stratospheric filaments in the troposphere, Journal of Geophysical Research: Atmospheres, 105, 10 169–10 184, 2000.

Kentarchos, A., Roelofs, G., and Lelieveld, J.: Simulation of extratropical synoptic-scale stratosphere-troposphere exchange using a coupled chemistry GCM: sensitivity to horizontal resolution, Journal of the atmospheric sciences, 57, 2824–2838, 2000.

Li, Q., Jacob, D. J., Logan, J. A., Bey, I., Yantosca, R. M., Liu, H., Martin, R. V., Fiore, A. M., Field, B. D., Duncan, B. N., et al.: A tropospheric ozone maximum over the Middle East, Geophysical research letters, 28, 3235–3238, 2001.

Lin, M., Fiore, A. M., Cooper, O. R., Horowitz, L. W., Langford, A. O., Levy, H., Johnson, B. J., Naik, V., Oltmans, S. J., and Senff, C. J.: Springtime high surface ozone events over the western United States: Quantifying the role of stratospheric intrusions, Journal of Geophysical Research: Atmospheres, 117, 2012.

Lin, M., Fiore, A. M., Horowitz, L. W., Langford, A. O., Oltmans, S. J., Tarasick, D., and Rieder, H. E.: Climate variability modulates western US ozone air quality in spring via deep stratospheric intrusions, Nature communications, 6, 2015.

Shapiro, M.: Turbulent mixing within tropopause folds as a mechanism for the exchange of chemical constituents between the stratosphere and troposphere, Journal of the Atmospheric Sciences, 37, 994–1004, 1980.

Zhang, L., Jacob, D. J., Yue, X., Downey, N., Wood, D., and Blewitt, D.: Sources contributing to background surface ozone in the US Intermountain West, Atmospheric Chemistry and Physics, 14, 5295–5309, 2014.

[revised manuscript text omitted]

Stratosphere-to-troposphere transport (STT) is considered a process of great importance for the EM region, as it influences tropospheric ozone levels during summer (Zanis et al., 2014).  Zanis et al. (2014) reported that STT processes feed stratospheric ozone into the upper troposphere, and subsequently the ozone-rich air masses are transported to the lower free-tropospheric levels through the characteristic strong summertime EMME subsidence. The main mechanism for STT events is tropopause folding (Stohl et al., 2003), developed by the ageostrophic flow in the jet stream entrance, which is associated with stratospheric intrusions into the troposphere (Danielsen and Mohnen, 1977). Tropopause folding events mainly occur at mid-latitudes and are characterized by tongues of anomalously high potential vorticity (PV), high ozone and low water vapor mixing ratios (Holton et al., 1995). Subsequent to transport into the troposphere, air with stratospheric origin is quasi-adiabatically stirred by large scale cyclonic and anticyclonic disturbances, which may lead to the formation of elongated streamers or isolated coherent structures. These can further dissipate and cascade down to smaller scales by non-conservative processes (such as radiative cooling/heating and turbulence), thus leading to irreversible mixing with the surrounding air (Shapiro, 1980; Appenzeller and Davies, 1992; Forster and Wirth, 2000).

Recently, Tyrlis et al. (2014) underscored the global hot spot of summertime fold activity between the EM and central Asia, in the vicinity of the subtropical jet, confirming the earlier findings of Sprenger et al. (2003). Moreover, they reported a striking dynamical link between fold activity over the EMME and the intensity of the South Asian Monsoon on interannual timescales. Convective activity over South Asia was found to regulate upper-level baroclinicity over the EMME and thus the fold occurrence over the region. In summary, the aforementioned studies show that STT events often occur in the EM region and in cases of deep stratospheric intrtusions can reach the lower troposphere (Zanis et al., 2003; Gerasopoulos et al., 2006; Akritidis et al., 2010).

Model sensitivity studies suggest that relatively high horizontal resolution might be beneficial for the representation of tropopause fold events and the associated intrusion of stratospheric ozone into the troposphere (Kentarchos et al., 2000). Lin et al. (2012) showed that a global high-resolution (50 km x 50 km) chemistry-climate model (GFDL AM3) captures the observed layered features and sharp ozone gradients of deep stratospheric intrusions. Moreover, Lin et al. (2015) carrying out sensitivity studies with the GFDL AM3 model, pointed out that using a finer horizontal resolution of 50 km x 50 km revealed an improvement in the reproduction of the day-to-day variability in the upper troposphere, as tropopause fold filamentary structures are better resolved in the finer model resolution simulations. Nevertheless, they suggested that the multidecadal hindcast simulations with the coarser resolution of 200 km x 200 km were also found suitable for quantifying the regional-scale interannual variability of stratospheric influence on lower tropospheric ozone over the western US.

[revised manuscript text omitted]

**3.2   The impact of tropopause folds on summertime tropospheric ozone**

The anomalies of O3 and O3s presented hereafter, are calculated as the differences between the average concentrations during fold events (average over 1866 timesteps) and the average concentrations during the remainder summer timesteps (average over 5864 timesteps). The role of tropopause folds in high tropospheric ozone levels during summer over the EMME is explored next. To this end, anomalies of the average ozone concentrations during fold events are constructed with respect to the concentrations during the rest of summer timesteps (Fig. 5, left). A distinct positive pattern is found in the middle troposphere (Fig. 5a, b and c) mainly over the EMME region, revealing an increase of ozone up to 7 ppb due to fold activity, which . An increase of ozone is also clear at 700 hPa (Fig. 5d) with mixing ratios of up to 4 ppb. During extreme events (above the 95th percentile of O3 concentrations during fold events), the range of O3 enhancement is found to be 19-33, 16-31, 17-24 and 11-19 ppb at 400, 500, 600 and 700 hPa respectively (Fig. S2 in the Supplementary Material).

The abovementioned enhancement in ozone levels is due to downward transport of ozone from the stratosphere through the folding process, as can be inferred from the respective anomalies for O3s shown in Figure 5 (right). Similar positive patterns, both quantitatively and spatially, are found for all examined pressure levels (Fig. 5e, f, g and h), supporting the hypothesis that the increase of tropospheric ozone during the selected fold events is mainly attributed to the transport of ozone of stratospheric origin. Analogous results with the same spatial features but less pronounced positive deviations were obtained by analyzing the anomalies of both O3 and O3s during the selected summer fold events with respect to their summer climatologies (not shown).

It should be mentioned that different definitions of stratospheric ozone tracer may have implications for the estimated stratospheric contribution, as has been pointed out by Zhang et al. (2014) . More specifically, a doubling of the diagnosed stratospheric ozone influence was found in GEOS-Chem simulations when the produced ozone in the troposphere, which was temporarily transported above the tropopause, was also considered as stratospheric (Lin et al. (2012) approach). The amount of ozone that is recirculated across the tropopause in the model depends on the vertical resolution. Thus, following the approach by Lin et al. (2012) is likely to yield an upper limit to the stratospheric contribution to tropospheric ozone in our results. Nevertheless, because the presently used middle atmosphere version of the EMAC model has relatively high resolution in the upper troposphere and lower stratosphere (about 500m), and because we initialize O3s well above the tropopause (100 hPa), we expect this effect to be small.

[revised manuscript text omitted]